# Effectiveness of Gold Nanorods of Different Sizes in Photothermal Therapy to Eliminate Melanoma and Glioblastoma Cells

**DOI:** 10.3390/ijms241713306

**Published:** 2023-08-27

**Authors:** Javier Domingo-Diez, Lilia Souiade, Vanesa Manzaneda-González, Marta Sánchez-Díez, Diego Megias, Andrés Guerrero-Martínez, Carmen Ramírez-Castillejo, Javier Serrano-Olmedo, Milagros Ramos-Gómez

**Affiliations:** 1Center for Biomedical Technology (CTB), Universidad Politécnica de Madrid (UPM), 28040 Madrid, Spain; javier.domingo@ctb.upm.es (J.D.-D.); marta.sanchez@ctb.upm.es (M.S.-D.); carmen.ramirez@ctb.upm.es (C.R.-C.);; 2Departamento de Química Física, Universidad Complutense de Madrid, Avenida Complutense s/n, 28040 Madrid, Spainaguerrero@quim.ucm.es (A.G.-M.); 3Grupo de Sistemas Complejos, Universidad Politécnica de Madrid, 28040 Madrid, Spain; 4Advanced Optical Microscopy Unit, UCCTs, Instituto de Salud Carlos III (ISCIII), 28222 Madrid, Spain; 5Departamento Biotecnología-B.V. ETSIAAB, Universidad Politécnica de Madrid, 28040 Madrid, Spain; 6Departamento de Oncología, Instituto de Investigación Sanitaria San Carlos (IdISSC), 28040 Madrid, Spain; 7Centro de Investigación Biomédica en Red para Bioingeniería, Biomateriales y Nanomedicina, Instituto de Salud Carlos III, 28029 Madrid, Spain; 8Departamento de Tecnología Fotónica y Bioingeniería, ETSI Telecomunicaciones, Universidad Politécnica de Madrid, 28040 Madrid, Spain; 9Experimental Neurology Unit, Center for Biomedical Technology, Universidad Politécnica de Madrid, Campus de Montegancedo s/n, Pozuelo de Alarcón, 28223 Madrid, Spain

**Keywords:** gold nanoparticles, photothermal therapy, melanoma, glioblastoma, cancer cells, apoptotic death, lysosomes

## Abstract

Gold nanorods are the most commonly used nanoparticles in photothermal therapy for cancer treatment due to their high efficiency in converting light into heat. This study aimed to investigate the efficacy of gold nanorods of different sizes (large and small) in eliminating two types of cancer cell: melanoma and glioblastoma cells. After establishing the optimal concentration of nanoparticles and determining the appropriate time and power of laser irradiation, photothermal therapy was applied to melanoma and glioblastoma cells, resulting in the highly efficient elimination of both cell types. The efficiency of the PTT was evaluated using several methods, including biochemical analysis, fluorescence microscopy, and flow cytometry. The dehydrogenase activity, as well as calcein-propidium iodide and Annexin V staining, were employed to determine the cell viability and the type of cell death triggered by the PTT. The melanoma cells exhibited greater resistance to photothermal therapy, but this resistance was overcome by irradiating cells at physiological temperatures. Our findings revealed that the predominant cell-death pathway activated by the photothermal therapy mediated by gold nanorods was apoptosis. This is advantageous as the presence of apoptotic cells can stimulate antitumoral immunity in vivo. Considering the high efficacy of these gold nanorods in photothermal therapy, large nanoparticles could be useful for biofunctionalization purposes. Large nanorods offer a greater surface area for attaching biomolecules, thereby promoting high sensitivity and specificity in recognizing target cancer cells. Additionally, large nanoparticles could also be beneficial for theranostic applications, involving both therapy and diagnosis, due to their superior detection sensitivity.

## 1. Introduction

Cancer is the leading cause of almost 10 million deaths worldwide. It is estimated that in the future, the number of new cancer cases worldwide will exceed the 19.3 million recorded in 2020, and that the number of deaths will exceed 10 million [1]. Traditional treatments for cancer include surgery, chemotherapy, and radiotherapy [2]. However, these treatments often lead to significant side effects. Due to the heterogeneous nature of cancer and the risks associated with current treatments, there is a need for the development of new therapeutic strategies. Innovative approaches, such as hyperthermia, are being explored and can be combined with conventional methods [3,4].

Nanoparticle-mediated photothermal therapy (PTT) is a non-invasive treatment that involves increasing the temperature of the target tissue to selectively kill cancer cells. This is achieved by using nanoparticles (NPs) as mediators that convert light into heat upon laser exposure. The elevated temperature in the tumor microenvironment directly affects cancer cells, triggering necrotic and apoptotic pathways. The extent of these pathways depends on the temperature reached during the irradiation procedure (typically ranging from 41–45 °C) [5]. Nanoparticles have a tendency to passively accumulate in tumors due to the enhanced permeability and retention effect caused by the leaky and disorganized tumor vasculature [6]. Furthermore, PTT promotes the dilation of leaky-tumor blood vessels, increasing their permeability and making tumor tissues more susceptible to temperature changes compared to healthy tissues [7,8]. Additionally, the thermal stress caused by PTT renders tumors more sensitive to conventional therapies, such as radiotherapy or chemotherapy [9,10].

The release of cytokines, chemokines and immunogenic intracellular molecules from cancer cells occurs due to the rupture of the cell membrane triggered by PTT. The release of these molecules can stimulate the immune system to eliminate cancer cells [10]. Additionally, PTT irradiation generates reactive oxygen species (ROS). Elevated levels of ROS can trigger protein denaturation, RNA/DNA destruction, and the activation of apoptosis. While PTT is currently effective in killing tumor cells, further research is needed to gain a deeper understanding of the mechanisms underlying cell death. Understanding the mechanisms of cell death induced by PTT is crucial for optimizing various parameters. These parameters include the dosages of NPs, the duration of laser irradiation, and the laser power, all of which need to be tailored to the specific type of cancer treated. By gaining a comprehensive understanding of the cell-death pathways and their modulation through PTT, it is possible to customize and optimize treatments for enhanced efficacy and minimal side effects [9,11].

The NPs used for PTT are typically photosensors or photoagents, usually made of iron oxide, graphene oxide and, most commonly, gold. Several shapes of NP are used in PTT, including spheres, stars, cages, flowers, or rods [12]. Gold nanorods (GNRs) are characterized by their remarkably high light-to-heat-conversion efficiency, making them the most commonly used photothermal agents in PTT. The optical properties of GNRs are influenced by their size and shape. They are governed by the phenomenon of localized surface plasmon resonance (LSPR), which occurs when GNRs are exposed to light radiation, causing the plasmon to oscillate on the surface of the GNR [13]. This distinctive characteristic of GNRs enables them to possess a remarkable capacity for photothermal conversion. In addition, GNRs can be biofunctionalized with biomolecules such as antibodies [14], peptides [15], proteins [16], or nucleic acids [17] to specifically target tumor cells. Another widely used molecule for GNR biofunctionalization is polyethylene glycol (PEG), which extends the blood-circulation time of GNRs and enhances their biocompatibility [18]. When GNRs are not biofunctionalized with PEG, their surfaces are highly recognizable by circulating blood-plasma proteins, which can alter the properties of GNRs. This recognition often leads to aggregation, rendering GNRs less effective in PTT [19]

Small GNRs (SGNRs), with approximate sizes of 40 × 10 nm, are the most commonly used GNRs for PTT. However, larger GNRs may possess better optical and thermal properties than smaller GNRs, making them useful for biological applications. Due to their increased optical cross sections, large GNRs (LGNRs) exhibit greater backscattering than conventional SGNRs. Additionally, their larger surface area allows their combination with a greater number of biomolecules, facilitating the specific targeting of cancer cells [20]. 

In the present study, we evaluate the effectiveness of two PEG-coated GNRs with different sizes in PTT and compare them to commercially available GNRs. The effects of LGNRs and SGNRs were evaluated on glioblastoma (CT2A) and melanoma (B16F10) cell lines undergoing PTT, while characterizing the phenomena of apoptosis and necrosis. 

## 2. Results

### 2.1. Temperature Curves

The efficacy of the synthesized large and small GNRs (LGNRs and SGNRs) in PTT was demonstrated by comparing them with commercial GNRs (CGNRs) measuring 10 × 41 nm. The results obtained with GNRs of the same size, SGNRs and CGNRs, were similar in all the subsequent sections, indicating that the outcomes achieved with the synthesized GNRs were highly comparable to those obtained with the commercial GNRs. The ability of the GNRs to convert light into heat was initially determined by analyzing the increase in temperature produced when increasing concentrations of the different types of GNR were irradiated with an 808-nanometer laser at 4.5 W for 10 min (Figure 1A). The LGNRs demonstrated the highest efficiency in converting laser light into heat at all the concentrations tested (Figure 1A). The LGNRs were able to produce an increase in temperature of 25.6 °C in response to the laser irradiation at the highest concentration tested (10 µg/mL) (Table 1). Moreover, even at 5 µg/mL, the LGNRs produced a greater temperature increase compared to the SGNRs and CGNRs at their highest concentration (10 µg/mL) (Figure 1A and Table 1). The smaller GNRs (SGNRs and CGNRs) showed similar temperature curves in response to the application of the laser (Figure 1A and Table 1). These results indicated that GNRs, especially the LGNRs, can be used at low concentrations (1–2 µg/mL) to produce hyperthermia conditions, and that the selected laser power (4.5 W) does not induce hyperthermia in the absence of GNRs. A negligible heating of only 2.2 °C was observed when the medium without GNRs was irradiated with the 808-nanometer laser for 10 min (Figure 1A).

For PTT applications, it is crucial to assess the photothermal stability of GNRs, as certain GNRs may undergo structural changes upon laser absorption [21]. To assess the photothermal stability of the GNRs and exclude morphological changes that could have produced shifts in the LSPR, the GNRs were subjected to three consecutive irradiation cycles to investigate whether high temperatures could have an impact on their structure and stability. The GNRs were tested at high concentrations (50 μg/mL) because the temperature reached after the repeated irradiation cycles at this concentration was higher than that obtained with the GNRs at lower concentrations, such as that used for the PTT treatments (2 μg/mL). This choice facilitated the easy observation of potential effects of repeated laser irradiation on the GNR structure. When the solution containing the GNRs was subjected to three laser-on–off cycles, the maximum temperatures reached in each cycle were similar for all the GNRs (Figure 1B). Only slight differences were observed in the cooling profiles of each type of particle. Therefore, the light-to-heat-conversion efficiencies were maintained over three consecutive cycles of heating and cooling, confirming the stability of the photothermal responses of all the types of GNR.

To assess the homogeneity of the temperature distribution in the medium, a GNR solution of 50 μg/mL was placed in the wells in which the cells were seeded for the in vitro PTT treatments. For these experiments, a higher concentration of GNRs (50 μg/mL) was chosen compared to that used for the PTT treatments (2 μg/mL) in order to clearly distinguish the increase in temperature produced in the well from the background temperature after the laser irradiation. The GNR solution was then irradiated with an 808-nanometer laser at 4.5 W for 30 s. Infrared camera images of the photothermal heating revealed that the entire well experienced uniform heating (Figure 1C). Hence, all the cells incubated with GNRs within the well were subjected to a uniform temperature increase during the in vitro PTT treatments. In addition, the control cells (the cells without GNRs) were not exposed to any alterations in temperature (Figure 1C).

### 2.2. Effects of Laser Irradiation and GNR Concentration on Cell Viability

In order to assess the impact of the application of the laser on the cell viability, the CT2A and B16F10 cells were exposed to laser irradiation for 10 min at several laser powers, ranging from 1 W to 4.5 W, in the absence of GNRs. The results showed that when the cells were irradiated at RT, none of the laser powers tested caused a statistically significant reduction in cell viability in either the CT2A or the B16F10 cell lines (Figure 2A). Dead cells were not observed using the calcein/IP assays 24 h after the laser irradiation at 4.5 W, although a lower number of cells was observed, probably due to a slowdown in the cell division (Appendix A). Therefore, a laser power of 4.5 W was selected to perform the in vitro hyperthermia experiments at RT. However, when laser irradiation was applied to the cells maintained at 37 °C to emulate physiological conditions, laser powers of 1.5 W and higher produced significant decreases in cell viability, particularly in the CT2A cells (Figure 2B and Appendix A). On the other hand, the B16F10 cells exhibited a significant reduction in viability when 2 W and above of laser power were applied (Figure 2B and Appendix A). Therefore, a laser power of 1 W was selected to perform the in vitro hyperthermia experiments at 37 °C, since the higher laser powers led to a significant decrease in cell viability.

To determine the impact of GNR concentration on cell viability and assess the cytocompatibility of the GNRs employed in this study, both the CT2A and the B16F10 cell lines were incubated with GNRs of different sizes at increasing concentrations (ranging from 1 to 5 µg/mL) for 24 h. The cell viability was then assessed using the XTT assay (Figure 3) and calcein/PI assays (Figure 3C,D and Appendix A). In the CT-2A cells, the GNRs did not cause a significant reduction in cell viability, even when used at a concentration of 2 μg/mL (Figure 3A). On the other hand, the B16F10 cells exhibited better tolerance to the presence of LGNRs, even at a concentration of 5 μg/mL, while the SGNRs and CGNRs at the highest concentration (5 μg/mL) significantly reduced the viability of these cells (Figure 3B). These results showed that when both cell lines were incubated with any GNR types up to a concentration of 2 µg/mL, the cell viability remained unaffected (Figure 3). Therefore, a GNR concentration of 2 μg/mL was selected as the optimal concentration for performing the in vitro hyperthermia treatments.

### 2.3. GNR Uptake by Cells

The internalization of GNRs into living cells has been extensively reported [22]. The uptake of GNRs by cells is dependent on several factors, such as size, shape, charge, surface chemistry, and cell type [23]. Thus, it is important to determine the uptake of each type of GNR in different cell lines. The ability of GNRs to be internalized by CT2A and B16F10 cells was assessed by confocal fluorescence microscopy (Figure 4) and dark-field microscopy (Figure 5) after exposing the CT2A and B16F10 cells to GNRs at 2 µg/mL for 24 h.

The presence of GNRs inside the cells was detected by confocal microscopy using the property of these GNRs to reflect the incident light (Appendix A). The intracellular presence of GNRs was confirmed by sectioning the confocal stack images along the XZ and YZ planes. The orthogonal projections made in an intermediate section of the cell body showed the presence of GNRs inside both cell types (the red spots in Figure 4), demonstrating that the LGNRs and SGNRs were internalized by both cell types.

To compare and quantify the uptakes of the different-sized GNRs in both cell lines, cells incubated with the LGNRs and SGNRs were observed using dark-field microscopy, as described in the Materials and Methods section. Due to their strong plasmon resonance, GNRs can be easily visualized with dark-field microscopy [24]. Dark-field images of both cell lines incubated with the LGNRs and SGNRs were captured, and the accumulations of the GNRs are shown as yellow/orange dots (Figure 5D–F,J–L). The results demonstrated that slightly more GNRs were bound to the B16F10 cells compared to the CT2A cells (Figure 5M). However, no significant differences were observed in the number of LGNRs and SGNRs bound to the B16F10 cells, indicating that both types of nanorod were able to bind to these cells in similar proportions. On the other hand, the SGNRs bound to the CT2A cells in a higher number than the LGNRs (Figure 5M). The control cells (the cells without GNRs) did not show any yellow/orange dots (Figure 5D,J).

### 2.4. Photothermal Therapy:Impact of Cell-Culture-Medium Temperature

Once the optimal concentrations of GNRs and laser power were determined, PTT was applied to the CT2A and B16F10 cells. The GNRs were incubated with the cells for 24 h. After the incubation period, the cells were rinsed with PBS to remove any excess or unbound GNRs to ensure that the experimental conditions mimicked the situation in vivo as closely as possible, with the GNRs exposed to cells and, subsequently, washed away by the surrounding fluid or systemic circulation. After the cells were incubated with the GNRs for 24 h and rinsed with the PBS, they were subjected to laser irradiation at a wavelength of 808 nm at 4.5 W for 10 min. This laser treatment was intended to induce PTT by utilizing the light-to-heat-conversion capability of the GNRs. The PTT treatments were performed at RT and at 37 °C to better simulate the physiological conditions and assess the impact of the PTT treatments in a relevant environment. After 24 h of laser irradiation, the cell viability was evaluated by XTT and Ca/IP. The results obtained when the PTT was applied at RT demonstrated that only the cells containing GNRs exhibited significant decreases in viability in response to the laser irradiation in both the CT2A (Figure 6A) and the B16F10 (Figure 6B) cell lines. Furthermore, the CT2A cells showed higher decreases in cell viability after the PTT compared to the B16F10 cells (Figure 6A and B, respectively). According to the results shown in Figure 6A,B, the LGNRs were slightly more efficient than the other types of GNR in reducing the viability of the CT2A and B16F10 cells when the PTT was applied at RT. In addition, neither the GNRs nor the laser alone produced a significant decrease in cell viability (Figure 6A,B). The results obtained when the PTT was applied at 37 °C were similar to those obtained at RT, but the cell-viability rates were even lower in both cell lines (Figure 6C,D). This PTT treatment applied at 37 °C resulted in the complete elimination of the cancer cells, especially in the B16F10 cells (Figure 6D), which exhibited a higher level of resistance to the PTT treatments applied at the RT (Figure 6B). Furthermore, in this case, neither the application of the laser nor the incubation with GNRs separately resulted in a decrease in cell viability (Figure 6C,D).

### 2.5. Cell-Death Mechanisms after PTT Treatments

After hyperthermia treatments, apoptosis and necrosis can occur sequentially or simultaneously, depending on the amount of NPs internalized by cells and the amount of energy dissipated by these NPs after irradiation [25]. It is important to note that necrosis often leads to the release of intracellular components into the extracellular medium, which can trigger inflammatory responses. Therefore, apoptotic cell death is preferred as a mechanism to eliminate cancer cells. 

The cell death produced in the CT2A cells preincubated with LGNRs and SGNRs 24 h after the application of the laser tended to occur through apoptosis rather than through necrosis (Figure 7A). Similar necrotic rates were observed in the CT2A cells 24 h after the laser irradiation in the presence of the LGNRs and SGNRs (Figure 7A). In the B16F10 cells, apoptosis was also found to be the predominant mechanism of cell death 24 h after the laser irradiation. The apoptotic pathway was the predominant method of eliminating these cells when using both types of GNR, although the LGNRs produced a slightly higher percentage of necrotic cells compared to the SGNRs (Figure 7B). 

There was a slight discrepancy in the number of live cells obtained from the XTT assay and flow-cytometry analysis in the B16F10 cells 24 h after the PTT treatments. The XTT assay appears to have yielded a higher rate of living cells (Figure 6B) compared to the flow cytometry (Figure 7B), which used apoptotic and necrotic markers. This inconsistency may have arisen from the use of different assays to evaluate the cell viability, with some early apoptotic cells (Anexin V^+^ cells) still able to reduce the XTT reagent, leading them to be classified as live cells in the XTT assay, while in the flow cytometry, these cells may have been stained with Annexin V. It is possible that the Annexin-V-positive cells still maintained their cell-membrane integrity and, thus, that the dehydrogenases required for the XTT reduction remained active within these cells, resulting in the classification of early apoptotic cells as live cells in the XTT assay. No significant differences were observed in the number of apoptotic cells when comparing the different types of GNR in both cell lines (Figure 7). The PTT treatments in the presence of the CGNRs yielded comparable rates of apoptotic cell death to those observed with the LGNRs and SGNRs, resulting in 49.7 ± 15.3% and 59.5 ± 10.2% of apoptotic cells in the CT2A and B16F10 cells, respectively.

To investigate the potential involvement of lysosomes in cell death following PTT treatments, lysosomal staining was performed using a fluorescent acidotropic probe (Lysotracker), which exhibits a high affinity for acidic organelles like lysosomes [26]. The evaluation of the lysosomal staining was performed 2 h after the PTT treatments on the CT2A and B16F10 cells to determine whether lysosomes play a significant role in the observed cell-death process.

Lysotracker is a fluorescent marker that specifically accumulates in intact lysosomes within live cells. When the lysosomal membrane is disrupted, a fluorescent lysosomal probe is released into the cytosol, resulting in a decrease in cellular-fluorescence intensity [26]. Figure 8 shows that in non-irradiated CT2A and B16F10 cells (cells incubated with GNRs but without PTT treatment), the lysosomal labeling appears to have been strong and punctuated in both cell lines (Figure 8A,E,I,M). However, when these cells were irradiated after the GNR incubation with the laser, the staining of the lysosomes 2 h after the PTT treatments became diffuse and significantly less intense (Figure 8B,F,J,N). Despite this change in lysosomal staining, the cells remained viable, as indicated by the positive calcein staining (Figure 8D,L,H,P). These results suggest that lysosome leakage is one of the biological events that occurs alongside cell death induced by PTT treatments, which is consistent with previous findings reported for other nanoparticle-based treatments [27].

## 3. Discussion

Localized physical treatments, such as nanoparticle-based PTT, utilize hyperthermia to induce damage and destroy cancer cells and tumor tissues. Unlike other methods, like chemotherapy, PTT does not typically lead to the development of resistance in cells [28] By using controlled heating via light-induced hyperthermia, NP-mediated-PTT can effectively target and eliminate cancer cells without the risk of the development of resistance. 

The use of GNRs is very common in various biomedical applications, such as drug delivery, bioimaging, and PTT. These GNRs possess optical properties that make them perfectly suitable for PTT due to their ability to convert near-infrared (NIR) light into heat, a phenomenon known as surface plasmon resonance (SPR). This characteristic is crucial in GNRs because they possess two specific peaks of plasmon absorption, which are determined by their size. The location of the SPR is dependent on the size of both dimensions of the GNRs [29]. Furthermore, GNRs have higher efficiency in photothermal conversion compared to other shapes of gold nanoparticle, such as nanospheres, which only exhibit a single plasmon band [30,31].

Furthermore, GNRs offer several advantages over nanoparticles of other shapes, including a longer systemic circulation time, increased retention time within the tumor, and superior targeting of tumor cells. However, in the context of recent advancements in device-mediated immunotherapy, spherical shapes may have an advantage, as they tend to accumulate more in immune organs, leading to enhanced delivery efficiency to immune cells [32]. 

In our case, the GNRs were found to be extremely efficient in the PTT treatments. The LGNRs proved to be a better option for elevating the temperature after the laser irradiation. They achieved an increase in temperature that was eight degrees higher than the SGNRs at the highest concentration (10 μg/mL). Even at non-cytotoxic concentrations like 2 μg/mL, the LGNRs produced an increase in temperature that was nearly six degrees higher compared to the SGNRs. Therefore, LGNRs are more efficient for PTT applications, as they can be used at lower concentrations. In addition, based on theoretical modeling, LGNRs are predicted to offer advantages in various biomedical imaging techniques. This is due to the fact that LGNRs have greater absorption and scattering cross-sections compared to the commonly used SGNRs [20]. As a result, LGNRs have the potential to provide enhanced imaging capabilities and improved signal detection in various biomedical imaging applications. In addition, the larger surface areas of LGNRs make them particularly attractive for biomedical applications. This increased surface area provides more sites for functionalization, allowing a higher number of biomolecules to be attached to these LGNRs [33]. This enhanced functionalization capability enables LGNRs to be conjugated with a greater number of biomolecules that specifically target certain types of cancer cell, thereby increasing the efficiency of PTT.

Additionally, we showed that the synthesized GNRs remained structurally intact even after the repeated application of the laser application (Figure 1B). This finding is important for potential future therapies that involve multiple rounds of laser treatment following GNR injections. It has been observed that high laser intensities can induce structural changes in irradiated GNRs, transforming them into shorter nanoparticles, or even spherical shapes, which can significantly alter their optical properties [25]. 

The assessment of GNR cytotoxicity is a crucial aspect of in vitro and in vivo PTT studies. The cytotoxicity of GNRs is influenced by the presence of hexadecyltrimethylammonium bromide (CTAB), which is used in the synthesis of GNRs [34,35]. However, when GNRs are modified with polyethylene glycol (PEG) on their surfaces, this leads to a reduction in cytotoxicity in in vitro experiments compared to non-PEGylated GNRs [36]. The coating of GNRs with PEG not only prevents direct contact with CTAB but also modifies the surface charge. The cytotoxicity of GNRs has been extensively studied in various human tumor cell lines, including Hep-2 [37] and A549 [38], as well as in rodent cell lines like CT2A [39] and B16F10 [40]. These studies provided valuable insights into the cytotoxic effects of GNRs on different cell lines. Indeed, when evaluating the cytotoxicity of GNRs, it is essential to consider both the concentration-dependent effects and the observed variations between cell lines. It is commonly observed that the cytotoxicity of GNRs increases with higher concentrations. However, it is important to note that the specific cell line tested can significantly influence the observed cytotoxic response, as demonstrated in this study. In the case of LGNRs, high concentrations may exhibit toxicity towards CT2A cells while being non-toxic towards B16F10 cells (as shown in Figure 3). It has been observed that GNR uptake and, therefore, GNR cytotoxicity, depends on the type of nanoparticle used, as well as on the cell type, which plays a role in the internalization process [41]. The surface chemistry of nanoparticles has a significant impact on cytotoxicity, while the morphology and size of nanoparticles only slightly affect cell viability [42]. 

In our case, all the tested GNRs had no effect on the cell viability when used up to 2 μg/mL. However, as the GNR concentration increased to 5 μg/mL, the CT2A cells tolerated the CGNRs better, while the viability of the melanoma cells was affected by the presence of small GNRs. This confirms that different cell types display dissimilar sensitivities to the presence of GNRs, with small GNRs being more cytotoxic to melanoma cells, which exhibit higher susceptibility to SGNRs and CGNRs compared to CT2A cells. Therefore, it is crucial to carefully analyze the concentration-dependent effects and consider the specific characteristics of different cell lines to accurately assess the cytotoxicity of GNRs. This comprehensive understanding aids in determining safe and effective concentrations for various applications and cell types. Regarding the toxicity of LGNRs, certain studies have indicated that these GNRs mainly accumulate in the liver and spleen, rather than in other organs, such as the heart, kidneys, and lungs, where SGNRs have been detected [43,44]. On the other hand, SGNRs accumulate in large amounts in tissues [44], although it should be noted that LGNRs are excreted at a slower rate in comparison to SGNRs [43].

Metal nanoparticles typically enter cells primarily through endocytosis [45], and their uptake is influenced by factors such as shape, size, surface charge, and cell type [46]. In this study, the LGNRs and SGNRs were internalized by both the glioblastoma and the melanoma cells, as demonstrated by the confocal fluorescence microscopy (Figure 4). It was observed that the binding and uptake of the GNRs exhibited slight variations, depending on the cell type (as shown in Figure 5). The B16F10 cells exhibited a slightly higher uptake of GNRs compared to the CT2A cells. However, no significant differences in uptake were observed between the LGNRs and the SGNRs in both cell lines. These results are consistent with previous findings, which described variations in the internalization ability of particles in different cell lines [47].

The results of the present study demonstrated that the PTT treatments applied at room temperature were effective in significantly eliminating both the CT2A and the B16F10 cells. Additionally, all the types of GNR were effective in the in vitro PTT treatments, although the LGNRs were only slightly superior in inducing cell death in both cell lines. This finding was consistent with the ability of LGNRs to generate a higher temperature increase upon laser irradiation. It is worth noting that the LGNRs produced a significantly higher temperature increase upon laser irradiation compared to the SGNRs, as shown in Figure 1A. However, they did not cause a significant reduction in cell viability compared to the SGNRs in any of the cell types (Figure 6). This discrepancy may be attributed to the temperature decrease observed when irradiating LGNRs in a cell-like medium, such as agarose, compared to an aqueous solution. The slight increase in cell mortality produced by LGNRs compared to SGNRs can therefore be attributed to the substantial shift in the SPR of LGNRs in a viscous medium, resulting in reduced absorption and, consequently, in reduced heat generation [48].

When comparing the residual living cells that remained alive after the PTT treatments, the results revealed that the B16F10 cells exhibited greater resistance to the hyperthermia treatments, with cell-viability rates above 25% for all the GNRs used. Conversely, the CT2A cells displayed lower cell-viability rates for all the tested GNRs. 

Melanoma is widely recognized as a tumor type that is highly resistant to cytotoxic agents due to its developed resistance to apoptosis [49], which is the main form of cell death induced by PTT treatments, as shown in Figure 7. However, the rapid increase in temperature provided by the laser application starting from 37 °C promoted the complete elimination of B16F10 cells, resulting in cell-death rates of nearly 100% (Figure 6D). Since the overexpression of several heat-shock proteins (HSP) was proposed as a plausible mechanism underlying the observed antitumor effects in hyperthermia treatments [50], it is possible that the temperature reached during the PTT treatment applied at 37 °C may play a role in altering the expression of these proteins. This alteration in HSP expression could contribute to a higher degree of cell death when cells are exposed to higher temperatures within shorter time frames.

It was found that melanoma-cell mortality decreased when applying increasing laser powers and using the same concentrations of GNRs [40]. In our case, the B16F10 cells were not completely eliminated when the laser was applied at 4 W at RT. Nevertheless, when the melanoma cells were irradiated at 37 °C, a temperature more representative of the in vivo cellular environment, it was possible to achieve the complete elimination of the melanoma cells. Remarkably, this outcome was achieved using the same concentration of GNRs and with a decreased laser power, of 1 W, a level closer to the value of the maximum recommended exposure under IR-light irradiation [51].

Regarding the use of PTT treatments to eliminate glioblastoma cells, there is significant variability in the results described in the literature due to the wide range of parameters that can be applied. These parameters include the nanoparticle concentration, laser power and the duration of its application, and the type of nanoparticle biofunctionalization. The cell-mortality rates using GNRs similar to the SGNRs described here increased to 80% when using a lower laser power (1.5 W), but with a high concentration of GNRs present in the cell-culture medium during laser irradiation at 808 nm [22]. Additionally, PTT using a low-power laser was tested with a high concentration of nanoparticles, resulting in a cell-mortality rate of 60% after a PTT treatment [52]. Other strategies include the biofunctionalization of GNRs to target glioblastoma cells, which allowed the achievement of cell-mortality rates ranging from 20% to 60% after PTT [15,53]. Therefore, the glioblastoma-cell-mortality rates obtained in this study after PTT using LGNRs and SGNRs at a low concentration (2 μg/mL) are superior to those used to eliminate glioblastoma cells using PTT described in the previous literature. Additionally, the application of PTT treatments at 37 °C, which closely resembles physiological conditions, allowed a reduction in laser power to 1 W, a value closer to that of low-power laser irradiation, which made it possible to obtain similar or even greater cell-death rates. This adjustment led to the near-complete elimination of the CT2A cells in this study.

Apoptosis and necrosis are the two most frequently observed cell-death pathways following PTT treatments. The processes of necrosis and apoptosis in response to PTT depend on the power of the laser and the temperature attained within the cancerous tissues during irradiation [5], with low-energy irradiation inducing apoptosis rather than necrosis. In necrosis, there is a loss of cell-membrane integrity, which leads to the uncontrolled leakage of cellular contents into the extracellular space. This leakage can induce inflammatory responses that have the potential to damage adjacent tissues and, potentially, promote secondary tumor growth. In contrast, apoptotic cells maintain their plasma-membrane integrity, thereby avoiding inflammatory damage to surrounding cells and tissues [54]. Furthermore, apoptosis can stimulate the immune response to inhibit the development of secondary tumor growth, since apoptotic cells usually activate antitumoral immunity through a process known as T-cell cross-priming [55]. Hence, the effectiveness of LGNRs and SGNRs in promoting apoptotic cell death rather than necrosis (as illustrated in Figure 7) demonstrated that these nanoparticles are optimal for PTT treatments. 

In addition, when apoptotic cells are not efficiently cleared (e.g., due to a high number of apoptotic cells), they can progress to secondary necrosis and lose their membrane integrity (PI^+^ cells). This process leads to the release of danger-associated molecular patterns (DAMPs), which are molecules that stimulate the immune system [56]. Considering that the cytometry analyses were performed 24 h after the PTT treatments, it is possible that some of the initial apoptotic cells underwent secondary necrosis. As a result, in an in vivo situation, these necrotic cells could potentially stimulate the immune system’s response to detect and eliminate cancer cells.

Previous studies demonstrated that GNRs can accumulate inside endosomes and lysosomes [57], and that lysosomal leakage may occur following PTT treatments [58]. The permeabilization of lysosomal membranes caused by PTT treatments resulted in the release of lysosomal enzymes into the cytosol, triggering both apoptotic and necrotic mechanisms of cell death. The rupture of lysosomal membranes occurred prior to cell death, as evidenced by a decrease in the lysosomal-marker staining while the cells were still viable (cells positive for calcein). This suggested that lysosomal-membrane leakage preceded plasma membrane rupture. These results are consistent with other studies that utilized nanoparticles for the elimination of cancer cells [26,27]. Therefore, targeting GNRs to lysosomes can potentially improve the outcomes of antitumor PTT treatments.

## 4. Materials and Methods

### 4.1. Synthesis and Characterization of GNRs 

The GNRs of different sizes with LSPR at 800 nm were prepared using a seeded-growth method with some modifications as described in [59]. 

Gold seeds. To prepare the growth solution, 9.111 g of CTAB (50 mM) and 870.5 mg (11 mM) of n-decanol were added to 500 mL of water and stirred at approximately 60 °C for 30–60 min. The mixture was then cooled down to 30 °C, and 250 µL of a 0.05 M HAuCl_4_ solution was added to 25 mL of the n-decanol/CTAB solution in a 50 mL glass beaker. The resulting mixture was stirred at 300 rpm for 5 min. Next, 125 µL of a 0.1 M ascorbic acid solution was added, causing the orange-yellow solution to slowly turn colorless. At this point, a freshly prepared 20 mM NaBH_4_ solution was injected (one shot) under stirring at 1000 rpm and 30 °C. The injection resulted in brownish-yellow solutions, and the seed solutions were aged for at least 60 min at 30 °C before use. It is important to note that the dimensions of the PTFE plain magnetic stirring bar (30 × 6 mm) used in the stirring process can strongly affect the quality of seeds.

Synthesis of anisotropic seeds. In a typical synthesis, 1000 μL of 0.05 M HAuCl_4_, 800 μL of 0.01 M AgNO_3_, 7 mL of 1 M HCl, and 1300 μL of 0.1 M ascorbic acid were added under vigorous stirring to 100 mL of a 50 mM CTAB and 13.5 mM n-decanol solution at exactly 25 °C. Once the solution became colorless, 6 mL of the seed solution was added under stirring, and the mixture was left undisturbed for at least 4 h. The solution changed from colorless to dark-brownish gray, and the recorded longitudinal LSPR was located at 725–730 nm. The small anisotropic seeds were centrifuged at 14,000–15,000 rpm for 60 min in 2 mL tubes. The precipitate was collected, redispersed with 10 mL of a 10 mM CTAB solution, and centrifuged twice under the same conditions. The final Au0 concentration was fixed to 4.65 mM (Abs400 nm: 10; optical path of 1 cm).

Growth of large GNRs (LGNRs). Briefly, in a typical synthesis, 3 mL of 0.01 M AgNO_3_, 1 mL of 0.05 M HauCl_4_, 3 mL of 1 M HCl, and 800 μL of a 0.1 M ascorbic acid solution were added under stirring to 100 mL of a 50 mM CTAB and 11 mM n-decanol solution at 35 °C. Next, 500 μL of the small anisotropic seed suspension was added under stirring. The mixture was left undisturbed for 4–6 h. The GNRs were forced to settle as sediment (by centrifugation at 8000 rpm, 30 min) to remove the excess of surfactant and redispersed in 10 mL of a 10 mM CTAB solution (GNR stock solution). This procedure was repeated twice to remove n-decanol traces. The resulting GNRs presented an average length of 68 ± 4 nm and diameter of 20 ± 2 nm (Appendix A). 

Growth of small GNRs (SGNRs). Briefly, in a typical synthesis, 3 mL of 0.01 M AgNO_3_, 1 mL of 0.05 M HauCl_4_, 3 mL of 1 M HCl, and 800 μL of a 0.1 M ascorbic acid solution were added under stirring to 100 mL of a 50 mM CTAB and 11 mM n-decanol solution at 35 °C. Next, 2500 μL of the small anisotropic seed suspension was added under stirring. The mixture was left undisturbed for 4–6 h. The GNRs were forced to settle as sediment (by centrifugation at 8000 rpm, 30 min) to remove the excess of surfactant and redispersed in 10 mL of a 10 mM CTAB solution (GNR stock solution). This procedure was repeated twice to remove n-decanol traces. The resulting GNRs presented an average length of 40 ± 2 nm and an average diameter of 10 ± 1 nm (Appendix A).

Functionalization of GNRs. Typically, PEG-SH (15 mg) was added under stirring to 5 mL of a freshly prepared aqueous suspension of GNRs (2 mM of Au0, 1 mM CTAB). After 1 h, the excess of free PEG-SH was removed by one centrifugation cycle (8000 rpm, 30 min). Next, the precipitate was redispersed in 5 mL of 10 mM buffer (EBSS).

Commercial GNRs (CGNRs) with an average length and diameter of 41 × 10 nm were obtained from Nanopartz (CP12-10-808-3KPA-PBS-50-1, Nanopartz Inc., Loveland, CO, USA), and were used for making some comparisons and to demonstrate the high quality of the synthesized GNRs.

### 4.2. Laser Irradiation

A continuous-wave laser operating at 808 nm with a maximum output power of 5 W (fiber-coupled laser system, HJ Optronics, Inc., San Jose, CA, USA) was vertically connected to a collimator lens (F-C5S3-780, Newport Corporation, Irvine, CA, USA) through a one-meter-long multimode optical fiber with a core diameter of 600 µm and a power-transmission efficiency of 90–99% (Changchun New Industries, Changchun, China). The collimator was fixed to irradiate the multiwell plates from the bottom in the in vitro studies. To determine the exact irradiation power, the fiber was connected to a power meter (PM USB LM-10, Coherent Inc., Santa Clara, CA, USA) using an SMA fiber adapter (PN 1098589, Coherent Inc., Santa Clara, CA, USA) and the software PowerMaxPC (Coherent Inc, Santa Clara, CA, USA).

### 4.3. Temperature Curves

The light-to-heat conversion efficiency of GNRs of different sizes (LGNRs, SGNRs and CGNRs) was determined by irradiating each type of GNR, resuspended in Dulbecco’s Modified Eagle Medium without phenol red (DMEM-PR), at increasing concentrations (1 to 10 μg/mL), with an 808 nm laser at 4.5 W for 10 min. A digital thermometer (Dostmann digital indicator model P795, Tempcontrol, NL, USA) with two temperature sensors (Pt-8330, Tempcontrol, NL, USA) and the software DE Graph (Tempcontrol, NL, USA) were used to obtain the corresponding heating curves during laser irradiation. One temperature sensor was placed in DMEM-PR at room temperature, while the other was placed in the GNR solutions, avoiding direct contact between the collimator and the sensor during irradiation. To determine the photothermal stability of the GNRs and exclude a decrease in their effectiveness, which could have prevented cell death during repeated laser treatments, each type of GNR at 50 μg/mL was subjected to three consecutives on–off irradiation cycles using the 808 nm laser at 4.5 W. The GNR samples were irradiated for 1 min, followed by switching off the laser, and allowing the solution to cool. This cycle was repeated three times.

### 4.4. Thermal Images

Thermal images were captured using a thermographic camera (RS PRO-RS 730, RS, London, UK) to demonstrate the uniform heating of a P96 multiwell plate containing GNRs in DMEM-PR at a concentration of 50 µg/mL. The samples were irradiated with an 808 nm laser at a power density of 4.5 W for 30 s. Images were taken every 10 s to monitor the temperature distribution across the well.

### 4.5. Cell Cultures

Murine glioma (CT2A) and murine melanoma (B16F10) cells were kindly donated by Prof. Ricardo Martínez and Prof. Lisardo Bosca, respectively. Both cell lines were maintained in DMEM (Gibco, Billings, MT, USA) supplemented with 10% heat-inactivated fetal bovine serum (FBS), 2 mM glutamine (Gibco, USA), 100 units/mL penicillin, and 100 µg/mL streptomycin. Cell lines were maintained at 37 °C in 5% CO_2_ and 95% air in a humidified atmosphere and passaged twice a week to ensure their optimal growth.

### 4.6. Cell-Viability Assays

To assess cell viability following GNR incubation and laser irradiation, two methods were employed: the XTT assay and calcein/propidium-iodide staining.

For XTT assay, the XTT kit (AppliChem Darmstadt, Germany) utilized the 2,3-bis-(2-methoxy-4-nitro-5-sulfophenyl)-2H-tetrazolium-5-carboxanilide salt (XTT), which was reduced by enzymes present in viable cells, resulting in the formation of a water-soluble dye [60]. The absorbance of each well was measured spectrophotometrically at 450 nm using an ELX808 microplate reader (BioTeK, Winooski, VT, USA). 

For calcein/propidium iodide (PI) dual-staining assay, a final concentration of 1 μM calcein (Invitrogen, Waltham, MA, USA, Molecular Probes, Eugene, OR, USA) was employed to stain living cells, producing green fluorescence, while 2 μM propidium iodide (PI) (Sigma-Aldrich, St. Louis, MO, USA) was utilized to stain dead cells, resulting in red fluorescence. These markers were added to each well and incubated at 37 °C for 20 min. Subsequently, fluorescence was evaluated using an inverted Leica DMI300 microscope equipped with a Leica DC100 digital camera (Leica, Nussloch, Germany).

### 4.7. GNR Uptake by Cells

The CT2A and B16F10 cells were seeded on a cell-treated glass coverslip with a diameter of 10 mm and placed into a well in a 24-well plate. Cells were seeded at a density of 5 × 10^4^ cells per well. Twenty-four hours after seeding, GNRs (50 µg/mL) were added to the cells and incubated for an additional 24 h. Cells were then rinsed in PBS 1× to remove unbound GNRs and fixed with 4% paraformaldehyde for 15 min at RT. Subsequently, the cells were stained with Phalloidin-FITC (1:1000; Sigma-Aldrich) for 3 h at RT in the dark. Nuclei were counterstained with Hoechst (1:500; Invitrogen). Coverslips were mounted on a glass slide with ProLongTM Glass Antifade Mountant (Life Technologies Corporation, Eugene, OR, USA). Images were captured using a confocal microscope (Laser Scanning Confocal Stellaris Falcon; Leica Microsystem, Wetzlar, Germany). Intracellular GNRs were visualized with a confocal reflectance mode. Using this mode, GNRs were detected by reflection under excitation with a laser line of 488 nm and by collecting the emission in the 480–500 nm range (Appendix A). To assess the internalization of GNRs into cells, orthogonal planes were captured at intervals of 40 µm. To obtain a global quantification of GNR uptake by cells, the same samples were examined using dark-field imaging. A CX43RF (Olympus, Tokyo, Japan) optical microscope equipped with a dark-field condenser was use for this purpose. Images of cells incubated with GNRs were obtained with a CMOS camera (EP-50, Olympus, Japan). Although individual GNRs cannot be visualized at this magnification, accumulations of GNRs were detected in both cell lines. The relative amounts of GNRs of different sizes bound to or internalized by the cells were quantified in pixels using the ImageJ software (NIH). Colored pixels above a baseline threshold were counted, as described in [23], allowing estimation of GNR uptake by the cells. This method was previously validated using inductively coupled plasma mass spectrometry [61].

### 4.8. Photothermal Therapy (PTT)

Firstly, the effects on cell viability of laser irradiation and GNR concentration were tested separately. The CT2A and B16F10 cells were seeded as explained previously and irradiated at RT for 10 min in the absence of GNRs to determine the effects of various laser powers (ranging from 0.5 W to 4.5 W) on cell viability. Cell lines were also irradiated at 37 °C for 10 min using laser powers ranging from 0.5 W to 2.5 W. Cell viability was assessed 24 h after laser irradiation, using the XTT and the calcein/PI assays. Similarly, the optimal concentration of GNRs was determined by testing concentrations ranging from 0 μg/mL to 5 μg/mL, in the absence of laser irradiation, and evaluating cell viability 24 h later.

For PTT experiments, CT2A and B16F10 cancer cells were seeded at 7 × 10^3^ cells/well in a p96 multiwell microplate (Fisher Scientific, Waltham, MA, USA), for a total volume of 120 µL. After 24 h, initial DMEM with 10% FBS was removed, and GNRs at a concentration of 2 µg/mL were added to the cells in DMEM-PR supplemented with 1% FBS. The cells were then incubated with the GNRs for an additional 24 h. Following the incubation period, the cells were washed with PBS to remove any GNRs that were not internalized by the cells. Cell viability was assessed using two different assays: XTT and calcein/PI. 

### 4.9. Quantification of Apoptosis by Flow Cytometry after PTT

The apoptosis induced in CT2A and B16F10 cells by PTT treatments was evaluated using flow-cytometry analysis at 24 h after the treatments. Annexin V (Bionova Científica) was used to stain apoptotic cells, while PI (Sigma-Aldrich) was used to stain necrotic cells. Cells were stained by adding Annexin V binding buffer and Cy5-conjugated AnnexinV according to the manufacturer’s instructions (Annexin V apoptosis-detection kit, ABCAM, Cambridge, UK). Cells were then harvested by tripsinization, centrifuged, and resuspended in Annexin V buffer. Next, PI (1 µg/mL) was added to the cells, which were then analyzed in a flow cytometer (FACSCalibur, BD Biosciences, San Jose, CA, USA), using the FL-2 channel for PI detection and the FL-4 channel for Cy5–Annexin-V detection. Annexin V binds to the membrane aminophospholipid phosphatidylserine, which is externalized from the inner to the outer leaflet of the plasma membrane in the early stages of apoptosis. When membrane integrity is lost, as seen in necrotic cell death, PI staining becomes positive. Therefore, apoptotic cells were defined as Annexin-V^+^/PI^−^ and necrotic cells as Annexin-V^+^/PI^+^ and Annexin-V^−^/PI^+^ cells. Ten thousand events were acquired and analyzed using the Flowjo software.

### 4.10. Lysosomal Staining

The CT2A and B16F10 cells were incubated with LGNRs and SGNRs at 2 µg/mL in DMEM-PR supplemented with 1% FBS for 24 h, and then washed to remove the excess GNRs, which were not internalized by the cells. The GNR-treated cells were then stained with Lysotracker Red (10 nM; Invitrogen) for 15 min and rinsed with DMEM-PR. Cells were then irradiated with an 808 nm laser at 4.5 W for 10 min and stained with calcein (1 µm) for 15 min. The integrity of lysosomes in calcein-positive living cells was assessed by analyzing the intensity of red Lysotracker fluorescence using an inverted Leica DMI300 microscope equipped with a Leica DC100 digital camera (Leica, Nussloch, Germany). The relative fluorescence intensity per cell obtained in the fluorescence images was quantified using the ImageJ software (NIH).

### 4.11. Statistical Analysis

The results are shown as the mean ± standard error of the mean from three to four experiments. The data were analyzed by single-factor analysis of variance followed by the post hoc Tukey’s honestly-significant-difference test. A significance level of *p* < 0.05 was chosen. GraphPad Prism version 9 (GraphPad Prism Software, San Diego, CA, USA) was used for all statistical tests.

## 5. Conclusions

In this study, we demonstrated the effectiveness of GNRs of two different sizes in promoting cell death through PTT in glioblastoma and melanoma cell lines. The use of GNR-based PTT treatments at a physiological temperature of 37 °C resulted in nearly complete efficacy, leading to the elimination of the glioblastoma cells and, notably, the melanoma cells. This is particularly significant considering that melanoma is known for its high resistance to conventional treatments, like chemotherapy. Moreover, we found that the primary cell-death pathway activated by these GNR-mediated PTT treatments is apoptosis, which is favorable, as the presence of apoptotic cells can stimulate antitumoral immunity in vivo. Furthermore, the calcein and lysosomal staining after the PTT treatments indicated that the rupture of lysosomal membranes took place prior to cell death. This suggests that the leakage of the lysosomal membranes preceded the rupture of the plasma membrane and might serve as one of the triggers of apoptosis-mediated cell death. Given the high efficacy of both types of GNR in PTT, LGNRs can be useful for biofunctionalization due to their greater surface area, which can be useful for attaching biomolecules and promoting high sensitivity and specificity in recognizing target cancer cells or tissues. Additionally, LGNRs could also be useful for theranostic (therapeutic and diagnostic) applications due to their superior detection sensitivity, providing superior contrast-enhanced imaging to smaller GNRs.

## Figures and Tables

**Figure 1 ijms-24-13306-f001:**
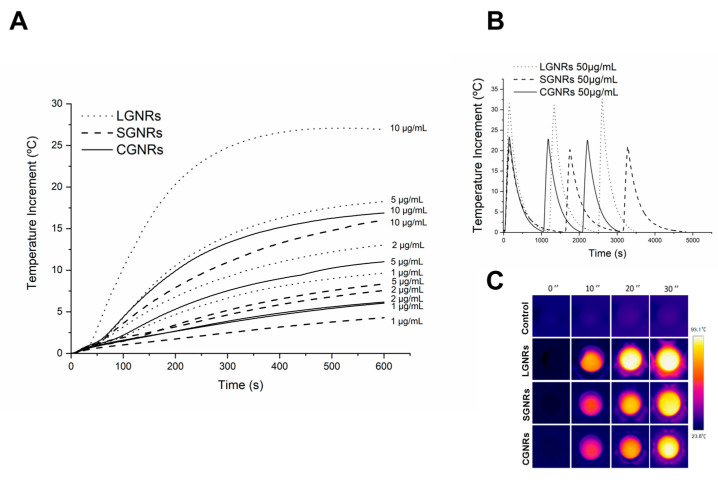
Temperature curves obtained by irradiating LGNRs, SGNRs, and CGNRs at increasing concentrations (1 µg/mL, 2 µg/mL, 5 µg/mL, and 10 µg/mL) for 10 min using an 808-nanometer laser at 4.5 W (**A**). Temperature curves obtained by irradiating LGNRs, SGNRs, and CGNRs at a concentration of 50 μg/mL showing three laser on/off cycles. The GNRs were irradiated for 60 s at 4.5 W, and then the laser was switched off until the sample returned to its initial temperature to start the next cycle of irradiation (**B**). Thermal images of LGNRs, SGNRs, and CGNRs in a multiwell p96 plate at a concentration of 50 µg/mL. The samples were irradiated at 4.5 W for 30 s and images were taken at 10-second intervals (**C**).

**Figure 2 ijms-24-13306-f002:**
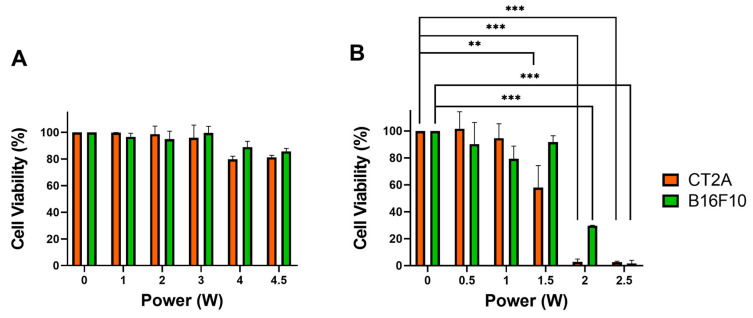
Viability of CT2A and B16F10 cells evaluated by the XTT test 24 h after irradiation with an 808-nanometer laser for 10 min at the indicated powers. Cells were irradiated at RT (**A**) and at 37 °C (**B**). ** *p* < 0.01; *** *p* < 0.001.

**Figure 3 ijms-24-13306-f003:**
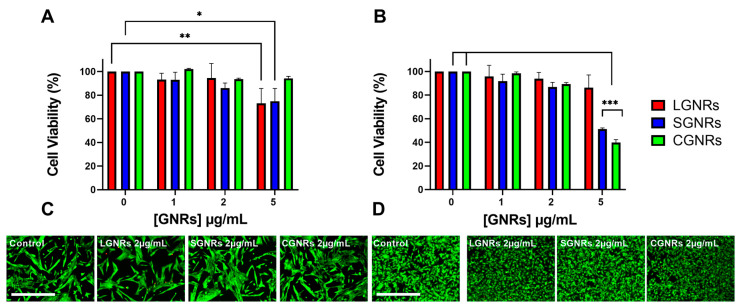
CT2A (**A**) and B16F10 (**B**) cell viability determined by the XTT assay after incubation with increasing concentrations of LGNRs, SGNRs and CGNRs for 24 h. Cell viability was also assessed by calcein/PI assay in CT2A (**C**) and B16F10 (**D**) cells incubated with all types of GNR at 2 µg/mL for 24 h. Viable cells are shown in green and dead cells in red. * *p* < 0.05; ** *p* < 0.01; *** *p* < 0.001. Scale bar: 400 µm.

**Figure 4 ijms-24-13306-f004:**
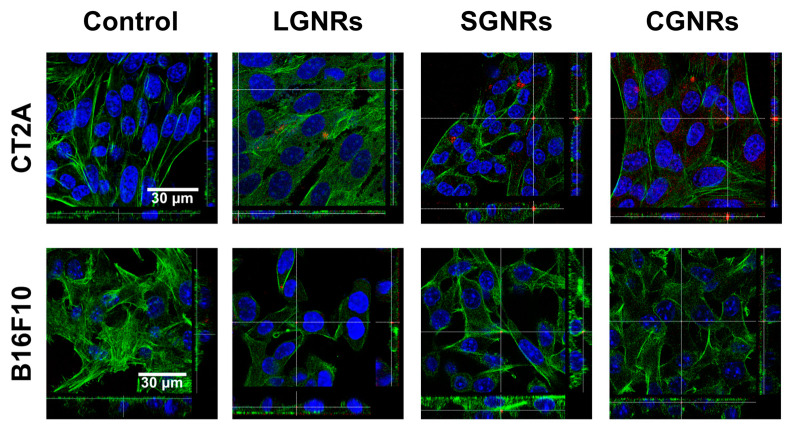
Confocal-laser-scanning-microscopy images of CT2A and B16F10 cells incubated with LGNRs, SGNRs, and CGNRs at 2 μg/mL for 24 h. After fixation, cells were stained with phalloidin (green) and the nuclei were counterstained with Hoechst (blue). The GNRs were detected by reflectance (red). Images show that all types of GNR (red) were effectively internalized into the cytosol of CT2A and B16F10 cells. Scale bar: 30 μm.

**Figure 5 ijms-24-13306-f005:**
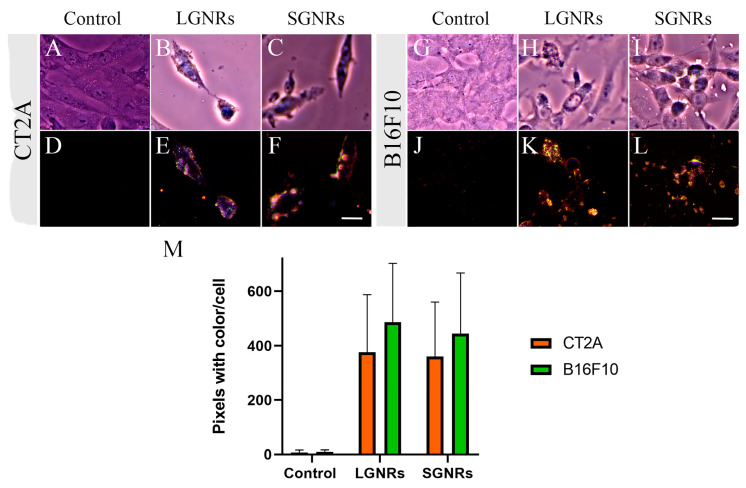
Bright field (**A**–**C** and **G**–**I**) and dark field (**D**–**F** and **J**–**L**) microscopy images of CT2A (**A**–**F**) and B16F10 (**G**–**L**) cells. Both cell types were incubated with LGNRs and SGNRs at 2 μg/mL for 24 h; control panels show cells without GNRs. Uptake of LGNRs and SGNRs by CT2A and B16F10 cells was quantified by counting colored pixels/cells (**M**) above threshold (control cells) using ImageJ software. Scale bars: 15 µm.

**Figure 6 ijms-24-13306-f006:**
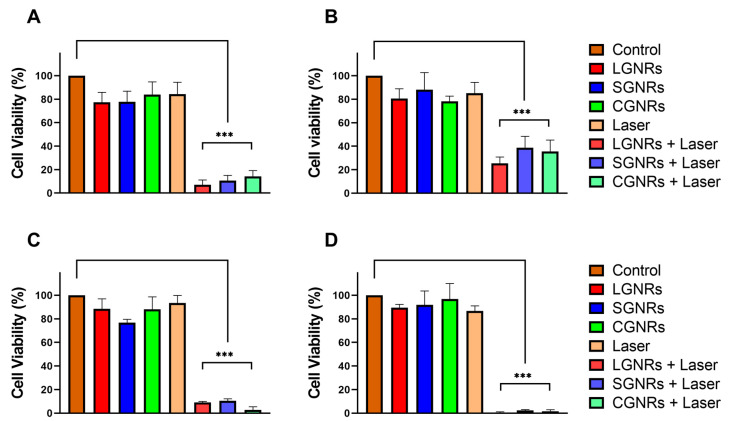
CT2A (**A**,**C**) and B16F10 (**B**,**D**) cell viability evaluated by the XTT assay after PTT treatments, cells preincubated with LGNRs, SGNRs, and CGNRs at 2 μg/mL for 24 h and then irradiated with an 808-nanometer laser at 4.5 W for 30 min at RT (**A**,**B**) and at 37 °C (**C**,**D**). *** *p* < 0.001.

**Figure 7 ijms-24-13306-f007:**
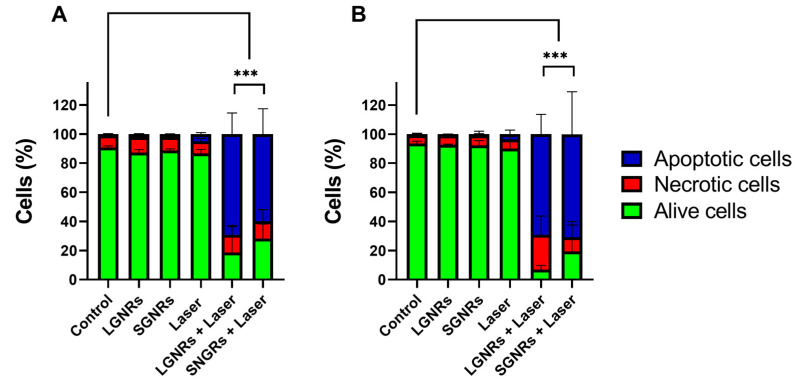
Analysis of cell death by flow cytometry (Annexin V and PI staining) in CT2A (**A**) and B16F10 (**B**) cells 24 h after PTT treatments. For PTT treatments, cells were preincubated with LGNRs and SGNRs at 2 μg/mL for 24 h and then irradiated with an 808-nanometer laser at 4.5 W for 30 min at RT. Apoptotic cells were defined as Annexin-V^+^/PI^−^ cells and necrotic cells as Annexin-V^−^/PI^+^ and Annexin-V^+^/PI^+^ cells. *** *p* < 0.001 (between apoptotic cell groups).

**Figure 8 ijms-24-13306-f008:**
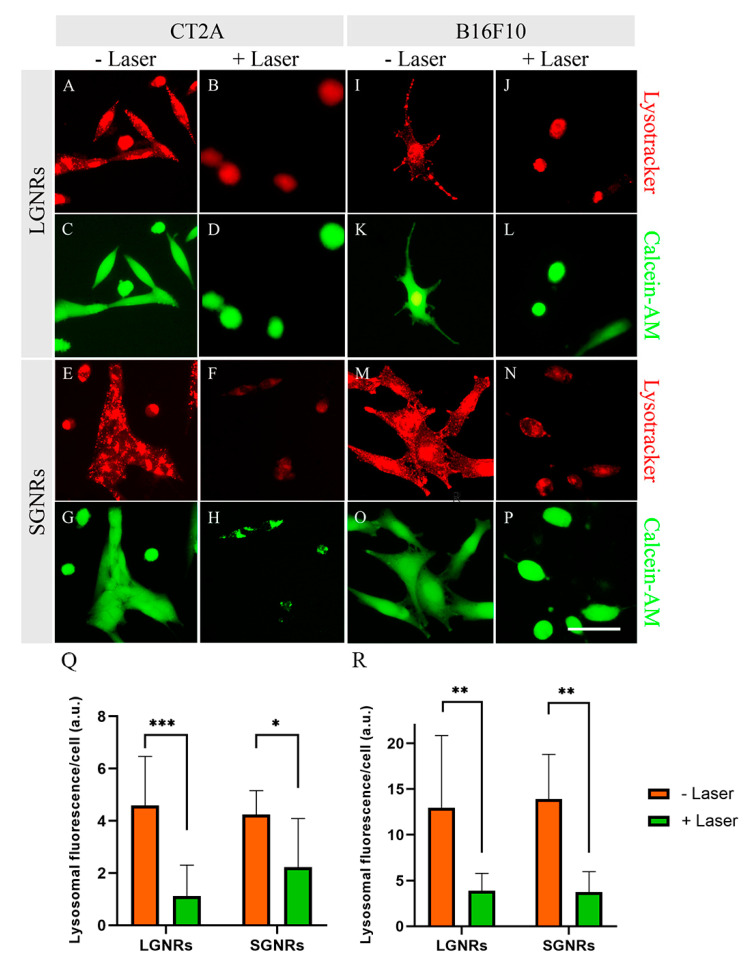
Lysosomal and calcein staining in CT2A (**A**–**H**) and B16F10 (**I**–**P**) cells incubated with LGNRs and SGNRs at 2 μg/mL for 24 h and then subjected to PTT treatments by irradiating cells with an 808-nanometer laser at 4.5 W for 30 min. Lysosomes were stained red with Lysotracker and viable cells were stained green with calcein*AM. Control cells: cells without GNRs. The fluorescence intensity of lysosomal staining per viable cell was quantified in all conditions using ImageJ in CT2A (**Q**) and B16F10 (**R**) cells. * *p* < 0.05; ** *p* < 0.01; *** *p* < 0.001. Scale bar: 20 μm.

**Table 1 ijms-24-13306-t001:** Temperature increments obtained when GNRs at the specified concentrations were exposed to an 808-nanometer laser at 4.5 W for 10 min.

Concentration (µg/mL)	Temperature Increment (°C)		
	CGNRs	SGNRs	LGNRs
10	17.5	17.7	25.6
5	12.0	9.8	18.9
2	7.3	8.7	14.5
1	7.2	5.5	10.7

## Data Availability

The data presented in this study are available on request from the corresponding author.

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
