# Peer review of "Effectiveness of Gold Nanorods of Different Sizes in Photothermal Therapy to Eliminate Melanoma and Glioblastoma Cells"

_ijms, 2023, doi:10.3390/ijms241713306_

Round 1
Reviewer 1 Report
The manuscript entitled “Effectiveness of gold nanorods of different sizes in photothermal therapy to eliminate melanoma and glioblastoma cells” by J. Domingo-Diez et al. addresses a remarkable issues in the use of plasmonic particles in oncology. The work is well designed and the manuscript is well presented and holds potential to make a significant impact among the readership of IJMS. However, prior to publication, I recommend minor revision.
The abstract lacks a description of the methods used in the study to detect changes in cell morphology and viability.
In the discussion, it is desirable to reflect whether there are data in the literature on the toxicity of large gold nanorods when used in vivo?
The conclusion does not fully reflect the work done by the authors. It is very short, I think it could be extended.
Author Response
Please find enclosed the file with the responses to the considerations of REVIEWER 1. We acknowledge the reviewers for their kind consideration, proposals for improvement, and valuable advice, which have helped improve the quality of the paper, presentation of results, and overall readability.
Please see the attachment.

Reviewer 2 Report
I have had the opportunity to review the manuscript titled "Effectiveness of gold nanorods of different sizes in photothermal therapy to eliminate melanoma and glioblastoma cells" submitted to “IJMS”. I must acknowledge that the problem addressed by the authors is indeed intriguing, and I appreciate the innovative approach they have taken to tackle it. However, I find myself not entirely satisfied with the work, as there are certain significant points related to typical photothermal treatment studies that this study does not cover. The paper lacks the required depth of conceptual analysis and coherent reasoning of the results; redundancy seems to be a recurring issue throughout the text.
Although I recognize the promising potential of the paper, I do have a number of concerns and questions that I believe need to be thoroughly addressed before we can proceed with the publication process. While the core idea is commendable, there are specific aspects that demand careful consideration to ensure both the robustness and clarity of the paper's content. Given the mentioned concerns, I would like to recommend that the manuscript undergo a comprehensive and major revision. I firmly believe that with the necessary improvements, the paper could truly shine and offer valuable insights to the readership.
1. To enhance the manuscript's structure, the Introduction and Results sections should be numbered as 1 and 2, respectively. By following this order, the Temperature curve at line 115 would appropriately be numbered as 2.1.
2. A discrepancy arises at line 125, where the highest temperature rise for LGNRs is mentioned as 25.8°C, whereas the table indicates 25.6°C. Please rectify this inconsistency for accuracy. Additionally, consider using decimal points instead of commas in numbers like 25.8°C to avoid confusion, and maintain uniformity throughout the text.
3. To accurately quantify cells, I recommend calculating micrograms of NPs per cell rather than pixels per cell or NPs per cell. This approach will provide a more meaningful representation of the relationship between NPs and cells.
4. At line 301, a typographical error is present. Kindly review and correct this mistake for clarity and precision.
5. Consider remaking Figure 7 using distinct colors such as black, red, and blue to enhance clarity, as the current version can be confusing.
6. The in-vitro biocompatibility of long, short, and commercial nanorods lacks clear explanation regarding how the length of gold nanorods influences cytotoxicity against the two cell lines.
7. Address the choice of 50 μg/ml for sample stability while conducting Photothermal Therapy (PTT) at 2 μg/ml. Additionally, provide the photothermal efficiency values at 2 μg/ml for the three types of nanorods to facilitate a direct efficiency comparison. It's advisable to perform homogeneity studies at the same concentration for consistency.
8. For more accurate quantification of cellular uptake, consider employing Inductively Coupled Plasma Mass Spectrometry (ICP-MS) alongside microscopy to gain a comprehensive understanding of uptake.
9. The paper lacks a comparison of apoptosis and necrosis concerning commercial nanorods. Highlight the advantages these materials offer over commercial nanorods in terms of these parameters.
10. While the targeting issue is mentioned, there is a lack of experimental evidence to support the targeting ability of the material. Consider conducting experiments to validate the targeting claim.
11. Improve the quality of figures by incorporating a broader spectrum of colors instead of solely relying on plain black, white, and gray. This addition of colors can enhance figure clarity and aid in better differentiation.
Author Response
Please find enclosed the file with the responses to the considerations of REVIEWER 2. We acknowledge the reviewers for their kind consideration, proposals for improvement, and valuable advice, which have helped improve the quality of the paper, presentation of results, and overall readability.
Please see the attachment.

Round 2
Reviewer 2 Report
Satisfied with the revised manuscript. The paper can proceed for publication.